# Impact Energy Dissipation and Quantitative Models of Injection Molded Short Fiber-Reinforced Thermoplastics

**DOI:** 10.3390/polym15214297

**Published:** 2023-11-01

**Authors:** Quan Jiang, Tetsuo Takayama, Akihiro Nishioka

**Affiliations:** Graduate School of Organic Materials Science, Yamagata University, 4-3-16 Jonan, Yonezawa 992-8510, Japan; t221291d@st.yamagata-u.ac.jp (Q.J.);

**Keywords:** impact strength, injection molding, interfacial shear strength, modeling, short fiber-reinforced thermoplastics

## Abstract

Glass short fiber-reinforced thermoplastics (SGFRTP) are used to reduce carbon dioxide emissions from transportation equipment, especially household vehicles. The mechanical properties required for SGFRTP include flexural strength, impact resistance, etc. In particular, impact resistance is an important indicator of the use of SGFRTP. For this study, a mechanical model was developed to explain the notched impact strength of SGFRTP injection molded products in terms of their interfacial shear strength. The values obtained from the model show good agreement with the experimentally obtained results (R^2^ > 0.95). Results also suggest that the model applies to different fiber orientation angle and a range of fiber lengths in the molded product that are sufficiently shorter than the critical fiber length.

## 1. Introduction

Reducing greenhouse gases such as carbon dioxide is recognized today as a common goal worldwide to help alleviate global warming. Particularly, the reduction in carbon dioxide emissions from transportation equipment such as automobiles and aircraft is required [1].

To reduce the carbon dioxide emissions of transportation equipment, the introduction of non-petroleum-based power systems is being considered in the automotive sector. One example is battery-powered electric vehicles. A battery is heavier than a conventional gasoline engine, thereby increasing the total vehicle weight. An increase in the vehicle weight also engenders increased collision energy. Concern exists that effects on the human body during accidents will be grievous. In the aircraft industry, carbon fiber-reinforced plastic (CFRP) has been applied to hulls to improve fuel efficiency through weight reduction [2]. For automobiles, CFRP has been used on the chassis of high-end cars, achieving a remarkable weight reduction of 100 kg. Suppose CFRP is applied to the chassis of regular passenger cars. In that case, it can be expected to contribute to carbon dioxide reduction significantly. However, this is expected to occur in the distant future because of the high costs of CFRP. To reduce vehicle weight to a practical level, the back door and instrument panel components can be assumed to be replaceable with even 3D-printed fiber-reinforced plastics [3]. This strategy has already been implemented for the back doors of some household automobiles. Currently, short glass fiber-reinforced thermoplastics (SGFRTP) containing more than 40 wt% discontinuous glass fiber (GF) are used to meet the required properties. GF has a higher specific gravity than carbon fiber (CF). Moreover, it requires higher fiber contents because of the discontinuous and random dispersion of the fibers. Therefore, even though the replacement of metallic materials with SGFRTP has been achieved, weight reduction effects have not been as significant as expected. In the case of SGFRTP, the molding is conducted mainly by injection molding, a mass production technique used for thermoplastics. In injection molding, the material is often supplied as pre-made composite pellets. Therefore, the fiber length cannot be greater than the pellet size. Moreover, the shear stress in the resin during the plasticization and inflow processes leads to fiber pressure loss. With injection molding, the fiber length in SGFRTP is invariably shorter than the pellet size. The fiber length in the SGFRTP injection molded products tends to be shorter than the critical fiber length [4]. Furthermore, with injection molding, the fibers flow into the mold cavity in a fountain flow pattern. Therefore, the fiber orientation varies in the thickness dimension.

The mechanical properties of SGFRTP injection molded products vary depending on the thickness and the molded product part. Usually, notched impact tests are used to evaluate the impact resistance of molded products. The impact resistance of SGFRTP injection molded products is also often evaluated according to the notched impact strength. The SGFRTP strength and impact resistance are governed by factors originating from the fiber, the matrix, and the interface between the fiber and the matrix [5]. Interfaces have been modeled in several ways, e.g., a narrow continuum region with graded properties, an infinitely thin surface separated by springs, and cohesive zones with specific traction–separation relations. Table 1 presents the dominant factors affecting the strength and impact resistance of SGFRTP. Among the factors originating from the fiber–matrix interface, interfacial shear strength (IFSS) and interfacial strength (IFS) have attracted attention as factors controlling the mechanical properties of fiber-reinforced plastics. 

In fact, pull-out [6], push-out [7], fragmentation [8,9], and pinhole pull-out methods [10] have been proposed to evaluate IFSS. The tensile test [11] has also been proposed to evaluate IFSS. Although these methods are fundamentally effective for capturing the fiber/matrix interface, they cannot be applied to actual molded products because they require the preparation of unique, molded products for evaluation. Given this background, Jiang et al. proposed a method to obtain the IFSS by short beam testing using SGFRTP injection molded products [12]. This method enables the evaluation of IFSS using molded products. The correlation between IFSS and mechanical properties such as strength and toughness can be derived quantitatively. In recent years, it has been reported that there is a positive qualitative correlation between IFSS and impact strength. 

T. K. Kallel et al. reported the effects of fiber orientation and interfacial shear strength on the mechanical and structural properties of GF-reinforced polypropylene (PP-g-MAH) and maleic anhydride (MAH) grafted SGFRTP [13]. The obtained results have shown that the interfacial adhesion was substantially improved when the PP-g-MA compatibilizer was added. The enhancement of the mechanical properties such as, impact strength and tensile strength, was affected by the increase in fiber orientation consistency and the enhancement of interfacial shear strength. However, the meso-mechanical model is only discussed for tensile strength.

Lou et al. reported the effects of some parameters, such as fiber length, fraction of GF, and fraction of maleic anhydride grafted PP-g-MAH on the mechanical properties and fracture structure of long fiber-reinforced thermoplastic composites (LFRT) [14]. The obtained results have shown that the mass fraction of GF is the main factor affecting the mechanical properties of the materials. Meanwhile, by studying the fracture mechanism of the 30 wt% GF LFRT, it can be concluded that the GF is subjected to most of the deformation force. When fracture occurs, the fiber is pulled out of the matrix in particular directions or broken. However, due to the inability to quantitatively analyze the failure state of long fibers, there was no discussion about quantitative models.

B.B. Yin et al. reported the effects of some parameters, such as fiber length and interfacial shear strength on the machine learning, and materials informatics approaches for evaluating the interfacial properties of fiber-reinforced composites [15]. The obtained results have shown positive correlations between IFSS and mechanical properties such as strength and toughness. X. Fang et al. reported positive correlations between interfacial shear strength and GF sheet-reinforced thermoplastics [16]. S. Parveen et al. even reported positive correlations between interfacial shear strength and glass fiber-reinforced epoxy composites [17].

From the above existing research, we can conclude that the correlation between IFSS and impact strength has not been clarified. Nevertheless, some correlations between impact properties and related parameters such as fiber length, fiber orientation, interfacial shear strength, and fiber volume fraction have been found. However, a quantitative model expressing impact strength using IFSS and IFS has not been proposed either. Both researchers and product development engineers want to have a simple universal model to predict impact strength. In our previous paper, we considered a quantification model based on pull-out [13]. However, the effects of fiber orientation angle, fiber length, etc., have not yet been discussed in detail. In particular, extreme fiber orientation angles, where all fibers are parallel to the fracture plane, need to be considered.

For this study, the correlation between the notched impact strength and the IFSS calculated using the short beam test was clarified for SGFRTP injection molded products. The model was validated by comparison with experimentally obtained results.

## 2. Materials and Methods

### 2.1. Materials

Polypropylene (PP, Novatec MA1B; Japan Polypropylene Corp. Tokyo, Japan, MFR = 21 g/10 min, 230 °C, 2.16 kg, *T*_m_ = 160 °C, *T*_g_ = 0 °C) and polystyrene (PS, Toyo Styrene GPPS G210C; Toyo-Styrene Co., Ltd. Tokyo, Japan MFR = 10 g/10 min, 230 °C, 2.16 kg, *T*_g_ = 105 °C) were used as matrices. Glass fibers (GF, ECS 03 T351; Nippon Electric Glass Co., Ltd. Tokyo, Japan, D = 13 μm, L = 3~5 mm) surface-modified with amino groups were used as fibers. Polypropylene maleic anhydride (MAH-PP, SCONA TSPP10213; BYK Additives & Instruments Co., Ltd., Wesel, Germany) and polystyrene maleic anhydride (MAH-PS, RESISFY R200; Denka Co., Ltd., Tokyo, Japan) were used as additives.

### 2.2. Sample Preparations

These materials were filled into a twin-screw extruder (IMC0-00; Imoto Machinery Co., Ltd., Kyoto, Japan) and were melt mixed at a 230 °C melting temperature and a 60 rpm screw speed. The configuration of the 15-mm-diameter screw installed in the extruder is portrayed in Figure 1a. The ratio of screw length to screw diameter is 25. 

To explore the impact of different fiber length distributions, PP, MAH-PP, and GF were filled into a single screw extruder (APEX JAPAN Co., Ltd., Saitama, Japan, AS-1) as portrayed in Figure 1b. All materials were melt mixed at a 200 °C melting temperature and a 20 rpm screw speed. In order to further control the variables, we added the same proportion of materials to the twin-screw extruder, and they were melt mixed at a 200 °C melting temperature and a 60 rpm screw speed. The melt-kneaded strands were pelletized using a pelletizer (cold-cut pelletizer; Toyo Seiki Co., Ltd., Tokyo, Japan) to obtain composite pellets with a 3 mm pellet length. The mixing ratios are shown in Table 2; we fixed the GF content at 30 wt% and MAH-PP and MAH-PS at 1.5 wt%, and used the twin-screw extruder at a 230 °C melting temperature (PP/GF, PP/MAH-PP/GF, PS/GF, PS/MAH-PS/GF). In addition, we fixed the GF content at 10 wt% and MAH-PP at 5 wt% using the single screw and twin-screw extruder at a 200 °C melting temperature (PP/MAH-PP/GF-Single screw (PP/MAH-PP/GF-S) and PS/MAH-PS/GF-Twin-screw (PP/MAH-PP/GF-T)).

The obtained composite material pellets were filled into an ultra-compact electric injection molding machine (C, Mobile0813; Shinko Sellbic Co., Ltd., Tokyo, Japan) and were injection molded to obtain beam-shaped molded products. This machine uses a pre-plunger system with a 10-mm-diameter plunger and mold clamping pressure of 29.4 kN. Table 3 shows the injection molding conditions. To clarify the effect of fiber orientation angle on impact performance, we designed two short beam-shaped specimens with different flow conditions, and a molded product thickness of 2 mm (specific size: 50 × 5 × 2 mm). These two types of products are shown in Figure 2. Figure 2a shows that beam specimens were performed in a single flow direction. Figure 2b shows that the beam with weld specimens were performed by adjusting the flow path so that a weld was formed in the center.

### 2.3. Charpy Impact Tests

A notch with a 0.25 mm tip radius was machined in the center of the obtained beam-shaped molded product. The notch depth was approximately 1 mm. The notched beams were subjected to Charpy impact tests using a Charpy impact tester (MYS-Tester Co., Ltd., Osaka, Japan). The loading speed was 2.91 m/s. The spun length was 40 mm. The Charpy impact strength of the specimens was found by calculating the absorbed energy *U* from the obtained swing angle based on the following Equation (1).
(1)$$a_{iN}=\frac{U}{B\left(W-a\right)}$$

Wherein, *B* stands for the molded product thickness. Also, *W* denotes the width of the molded product and *a* represents the notch depth.

### 2.4. Short Beam Share Testing for Determination of IFSS

The obtained beams were subjected to short beam share testing using a small universal mechanical testing machine (MCT-2150; A&D Co., Ltd., Tokyo, Japan). The loading speed was 10 mm/min and the spun length was 10 mm. The obtained load–deflection curve was differentiated by deflection to obtain the stiffness–deflection curve. Figure 3 presents an example of the stiffness–deflection curve. In the case of SGFRTP injection molded products, the fibers in the molded products are oriented randomly. In other words, most of the fibers are oriented obliquely to the flow direction. However, when a three-point bending load is applied, a bending moment and a shear stress are generated at the loading plane. The shear stresses are conjugate. They reach their maximum at the neutral plane. In the short beam test, high shear stress is generated near the neutral plane by shortening the spun length, which induces slippage at the interface. In the case of obliquely oriented fibers, the interface slippage is regarded as occurring first in either the parallel or perpendicular direction to the loading direction and then in the opposite direction as loading increases. In other words, a discontinuous decrease in stiffness is observed at two points. The point at which the stiffness decrease occurs under small shear stress is designated as Point 1. The point at which the decrease occurs under large shear stress is called Point 2.

In the case of SGFRTP injection molded products, the fibers near the neutral plane are oriented at an angle close to perpendicular to the flow direction. Therefore, the interface slippage is expected to occur first in a direction parallel to the loading direction. In other words, the relation between IFSS and shear stress at Point 1, in this case, can be expressed by Equation (2) below.
(2)$$IFSS=\frac{\tau_{1}}{\cos\theta_{f}}=\frac{3P_{1}}{4BW\cos\theta_{f}}$$

Wherein, *θ_f_* is the orientation angle of the GF dispersed near the neutral plane. The same relation at Point 2 can be expressed as the following Equation (3).
(3)$$IFSS=\frac{\tau_{2}}{\sin\theta_{f}}=\frac{3P_{2}}{4BW\sin\theta_{f}}$$

From Equations (2) and (3), *θ_f_* is expressed by Equation (4) below.
(4)$$\theta_{f}=\tan^{-1}\frac{\tau_{2}}{\tau_{1}}=\tan^{-1}\frac{P_{2}}{P_{1}}$$

Jiang et al. showed that this angle represents a frequent orientation angle in molded products [12]. The IFSS is obtainable by substituting the *θ_f_* obtained in Equation (4) into Equations (2) or (3). In this study, the *IFSS* was calculated based on the abovementioned method. The correlation between the *IFSS* and the Charpy impact strength was investigated. In this study, all *IFSS* evaluations were conducted using beam specimens without weld.

### 2.5. Tensile Testing for Determination of IFS

The obtained beam with weld specimens was subjected to tensile testing using a small universal testing machine (IMADA Co., Ltd., Toyohashi, Japan, FSA-1KE-1000N-L). The loading speed was 0.5 mm/min; the span distance was 22 mm. The *IFS* of the specimens was found by using the maximum load *P_max_* based on the following Equation (5) [11].
(5)$$IFS=\frac{P_{max}}{BW}$$

In this study, the *IFS* was calculated based on the method described above and all *IFS* evaluations were conducted using beam with weld specimens. The correlation between the *IFS* and the Charpy impact strength was also investigated.

### 2.6. Fiber Length Measurement

To measure the fiber length in the molded product, the area subjected to the Charpy impact test was cut out by machining. The cut-out area was placed in a small electric furnace (FT-HP-100; Full-Tech Co., Ltd., Osaka, Japan) and was fired at 600 °C for 4 h to obtain the residue. Then, the residue was spread out on a glass slide. Details of the residue were photographed using a phase contrast microscope (BA410EPH-1080; Shimadzu Rika Corp., Tokyo, Japan). Then, using image analysis software (WinLOOF ver.7.0; Mitani Corp., Fukui, Japan), the fiber lengths of more than 500 glass fibers in the photographs were measured. The average fiber length and its standard deviation were found.

### 2.7. Fiber Orientation Measurement

The fiber orientation corresponding to the core layer in the area subjected to the Charpy impact test was photographed using a microfocus X-ray CT system (ScanXmate-D225RSS270; Comscantecno Co., Ltd., Kanagawa, Japan) and a sub-microfocus X-ray CT unit (MARS TOHKEN SOLUTION Co., Ltd., Tokyo, Japan, TUX-3200N). The obtained fiber orientation photographs were used to ascertain the average fiber orientation angle of more than 350 fibers with respect to the flow direction using image analysis software (WinLOOF ver.7.0; Mitani Corp., Fukui, Japan).

### 2.8. Fracture Surface Observations

The fracture surface obtained after the Charpy impact test was photographed using a phase contrast microscope and a digital microscope (TG200HD2-Me; Shodensha Co., Ltd., Osaka, Japan). The photographic angle was set as 0° for the phase contrast microscope and 45° for the digital microscope, with respect to the side of the specimen. Using image analysis software, the fiber pull-out lengths of more than 300 fibers were measured from the photographs taken using a phase contrast microscope. The average pull-out length and its standard deviation were calculated.

## 3. Results

### 3.1. IFSS of Injection Molded SGFRTP

Figure 4a presents the results of using the twin-screw extruder at a melting temperature of 230 °C for IFSS evaluation. The error bars in the figure represent the standard deviation. Results showed a decreasing trend with increasing fiber content. However, the IFSS tended to increase with the addition of maleic anhydride-modified polymer, which is compatible with the matrix. This trend is consistent with the results reported by Kallel et al. for PP/GF-30 wt% [13].

Figure 4b presents the results of using the twin-screw and single-screw extruders at a melting temperature of 200 °C for IFSS and IFS evaluation. The error bars in the figure represent the standard deviation. Results showed that the results of different extruders are almost the same.

### 3.2. Charpy Impact Strength of Injection Molded SGFRTP

Figure 5a portrays the results of using the twin-screw extruder at a melting temperature of 230 °C for evaluation of the Charpy impact strength. The error bars in the figure indicate the standard deviation. The value of *a*_iN_ tended to increase with the increase in fiber content. Furthermore, the addition of maleic anhydride-modified polymer, which is compatible with the matrix, showed a tendency to increase the value. This trend is consistent with results reported by Kallel et al. for PP/GF-30 wt% [13].

Figure 5b portrays results of using the twin-screw and single-screw extruders at a melting temperature of 200 °C for evaluation of the Charpy impact strength. The error bars in the figure indicate the standard deviation. The value of *a_iN_* tended to increase obviously when using the single-screw extruder. Furthermore, the beam with weld specimen with the same orientation angle as the impact direction (perpendicular to the flow direction of injection molding) showed almost the same *a_iN_* results for different extruders.

## 4. Discussion

### 4.1. Quantitative Models of Energy Dissipation at Impact Loading of Beam Specimens’ SGFRTP

Figure 6 presents results of phase contrast microscope observations of the notch tip area. As these photographs show, fiber pull-out was observed near the notch tip in all specimens of the compositions examined for this study. As the fiber content increased, the number of pulled-out fibers tended to increase. Figure 7 presents results of digital microscopic observations. These photographs were taken with magnification that accommodates viewing the entire fractured surface. The red dotted wireframe emphasizes the entire fracture surface inside. These photographs indicate that lines are drawn through the entire fractured surface of all the specimens of the compositions examined for this study. These results suggest that the main mechanism of impact energy dissipation in the studied compositions is fiber pull-out. This impact fracture behavior is consistent with the results reported by Quan et al. [18].

Figure 8 shows the average fiber pull-out length, *l_p_*, as measured from the phase contrast microscope observation. The error bars in the figure show the standard deviation. Actually, *l_p_* tended to become shorter with increasing fiber content and interfacial shear strength. The *l_p_* of PS was shorter than that of PP compared to that of matrix. 

Figure 9 shows the fiber content dependence of the average fiber length of the fibers, *L_f_*, present in the molded product. The error bars in the figure show the standard deviation. Results show that *L_f_* tends to become shorter as the fiber content increases. The *L_f_* of PS was shorter than that of PP when compared with that of matrix. At the same time, we should also note that compared to the unadded matrix, the addition of MAH-PP or MAH-PS affected the fiber length and the pull-out fiber’s length. With the addition of MAH-PP or MAH-PS, the fiber length was slightly shorter compared to that without addition. Results indicate that an increase in interfacial shear strength will cause damage to the fibers during injection molding. The relation between *L_f_* and *l_p_* in Figure 8 and Figure 9 suggests that some correlation exists between *L_f_* and *l_p_*. Figure 10 presents a relation between *L_f_* and *l_p_*, approximated by the linear function equation shown in the figure. When the dispersed fibers are shorter than the critical fiber length *l_c_* shown in Equation (6) below, fiber pull-out dominates the yield condition of SFRTP.
(6)$$l_{c}=\frac{\sigma_{F}d}{2\tau }$$

In that equation, *d* is the fiber diameter and τ denotes the IFSS. The shear stress inside the fiber is maximal at half of the fiber length. In other words, theoretically, a 2:1 relation can be found between *L_f_* and *l_p_* [19]. The relation between *L_f_* and *l_p_* obtained in this study is approximately 2:1, which is equal to the theoretical value. This result also indicates that the addition of MAH-PP or MAH-PS does not affect fiber pull-out. This finding implies that the energy dissipation quantitative models in the Charpy impact test are mostly attributable to fiber pull-out. Figure 11 shows the value of *L_f_* divided by *l_c_*. This value is smaller than 1, which means that the yield condition of SFRTP is dominated by fiber pull-out. All results in the figure are less than 0.5. From these results, one can infer that the energy dissipation quantitative models in the Charpy impact test of the compositions studied in this paper are attributable to fiber pull-out.

### 4.2. Notched Impact Strength of Fiber Pull-Out

Based on the discussion presented above, we assume for these analyses that most of the energy dissipated in the Charpy impact test comes from fiber pull-out. The amount of energy dissipated by this mechanism is modeled using IFSS. Figure 12 shows the energy dissipation region during the Charpy impact test. The energy dissipation attributable to fiber pull-out occurs only in the region where the fiber is pulled out. Therefore, the energy dissipating region *V_P_* is expressed using Equation (7).
(7)$$V_{P}=l_{p}B\left(W-a\right)$$

The energy dissipated by the fiber withdrawal is regarded as coming from the friction at the interface between the fiber and the matrix phase. Therefore, the total amount of energy dissipated *U_P_* can be expressed by Equation (8).
(8)$$U_{P}=\tau \cos\varphi l_{p}S_{f}V_{P}$$

In that equation, *φ* denotes the orientation angle of the fiber with respect to the loading direction. In the Charpy impact test, the Charpy impact strength is obtained by dividing *U_P_* by the ligament area. Therefore, the final theoretical value *a_Model-P_* is expressed by the following Equation (9).
(9)$$a_{Model-P}=\frac{U_{P}}{B\left(W-a\right)}=\tau \cos\varphi S_{f}l_{p}^{2}$$

### 4.3. Quantitative Models of Energy Dissipation at Impact Loading of Beam with Weld Specimens’ SGFRTPs

Figure 13 presents results of phase contrast microscopy observations of the notch tip area. As shown in these photographs, fiber pull-out was observed near the notch tip in all beam specimens. Compared to the beam specimens, no fiber pull-out was observed in the Charpy impact test beam specimens with weld. Figure 14 presents results of digital microscopy observations. These photographs were taken at a magnification that allows the entire fracture surface to be viewed. These photographs show that lines are drawn through the entire fractured surface of beam specimens. Compared to the beam specimens, almost no lines are drawn through the fractured surfaces of beam with weld specimens. These results suggest that interfacial debonding is the main mechanism of impact energy dissipation in the beam with weld specimens.

Based on the discussion presented above, we assume for these analyses that most of the energy dissipated in the Charpy impact test comes from interfacial debonding when the fiber orientation direction is the same as the impact direction. Figure 15 shows the area of energy dissipation during the Charpy impact test. The energy dissipation due to interfacial debonding occurs only in the region of the fiber-to-fiber distance 〈*L_T_*〉 between the welding point. Therefore, the energy dissipating area *V_D_* is expressed by the following Equation (10).
(10)$$V_{D}=\langle L_{T}\rangle B\left(W-a\right)$$

The energy dissipated by the interfacial debonding is considered to come from the IFS between the fiber and the matrix phase. Therefore, the total amount of energy dissipated *U_D_* can be expressed by Equation (11).
(11)$$U_{D}=IFS\cdot V_{D}$$

In the Charpy impact test, the Charpy impact strength is obtained by dividing *U_D_* by the ligament area. Therefore, the final theoretical value *a_Model-D_* is expressed by the following Equation (12).
(12)$$a_{Model-D}=\frac{U_{D}}{B\left(W-a\right)}=IFS\cdot \langle L_{T}\rangle$$

### 4.4. Validity of the Proposed Theory

To perform calculations using these models, *φ* and 〈*L_T_*〉 must be found. Figure 16 presents X-ray CT observations taken from the surface of the molded product to about 1 mm in the thickness direction using a twin-screw extruder with a melting temperature of 230 °C for injection molded beam specimens. Since notches are introduced in the width direction in this study, we extracted more than 350 fibers from the region excluding the notches. We obtained the average orientation angle and its distribution. Figure 17 presents the orientation angle distribution and the average orientation angle. In all compositions, the fibers were mostly oriented perpendicular to the flow direction. As described in this paper, *φ* was used to obtain *a_Model-P_* using the average orientation angle. Figure 18 presents X-ray CT observations taken from the surface of the molded product to about 1 mm in the thickness direction using a twin-screw and single-screw extruder with a 200 °C melting temperature beam and a beam with weld specimens. We obtained the average orientation angle using the same method as in Section 4.1. We also obtained the average fiber-to-fiber distance near the welding point region, excluding the notches. As described in this paper, 〈*L_T_*〉 was used to obtain *a_Model-D_* using the fiber-to-fiber distance.

Figure 19 shows the correlation between *a_iN_*, *a_Model-P_*, and *a_Model-D_*. The correlation of SGFRTPs shown in this figure is proportional. The coefficient of determination *R*^2^ is close to 0.95, which is a good agreement. Also, the verification results show that the impact strength can be explained by Equation (9) when fiber pull-out is the main source of energy dissipation in quantitative models. In addition, the impact strength can be explained by Equation (12) when interfacial debonding is the main source of energy dissipation in quantitative models.

It is important to note the correlation between the interfacial debonding model and the cohesive force model (CFM) in this study. The cohesive zone approach is used to describe fracture and failure behavior in various material systems [20,21,22]. In the CFM, a stress-based criterion for debonding and a frictional resistance-based criterion for interfacial sliding were used to capture debonding and sliding [23]. In the Charpy impact test, debonding is postulated to occur under the combined action of normal tensile stress (mode I) at the interface. The physical meaning of 〈*L_T_*〉 can have the same meaning as crack tip opening displacement (COD). The same physical meaning can also illustrate the reliability of this model.

## 5. Conclusions

This study investigated the correlation between IFSS, IFS, and impact strength of injection molded glass fiber-reinforced thermoplastics made of a polypropylene and polystyrene matrix. 

►IFSS decreased with the increase in fiber content.►IFSS and IFS showed almost no changes following change in fiber length.►Charpy impact strengths increased with the increase in fiber content or fiber length.►Charpy impact strengths showed almost no changes when the fiber orientation direction was the same as the impact direction. ►The relation between *L_f_* and *l_p_* obtained in this study was approximately 2:1, which is equal to the theoretical value. At the same time, both are shorter than the critical fiber length. Usually when the fiber is shorter than the critical fiber length, the fiber break does not happen in SGFRTP.►Fracture surface observation results indicated that fiber pull-out and interfacial debonding were the primary sources of energy dissipation in the quantitative models.

Two theoretical equations with fiber pull-out and interfacial debonding as the main energy dissipation sources in the quantitative models were developed and correlated with the experimental results. Good agreement was obtained (R^2^ > 0.95). Furthermore, the fiber orientation at the interface will not affect the universality of the model.

This study clarified the quantitative models for interfacial shear strength, interfacial strength, and impact strength of injection molded beam-shaped specimens with different fiber orientation angles. However, this is based on the situation where the dispersed fiber length in injection molded products is less than the critical fiber length. When the fiber length distribution is longer than the critical fiber length in the molded product, especially in the fiber extraction model, the relationship between the fiber extraction length and the critical fiber length needs to be further clarified. Since fibers with weaker interfacial strength or less susceptibility to fracture during molding, such as organic fibers, plant fibers, etc., can be used, the universality of this model needs to be further verified.

## Figures and Tables

**Figure 1 polymers-15-04297-f001:**
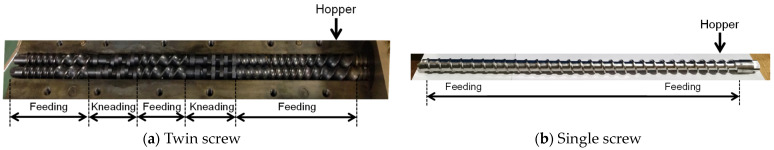
Screw configuration used for the twin and single screw extruder.

**Figure 2 polymers-15-04297-f002:**
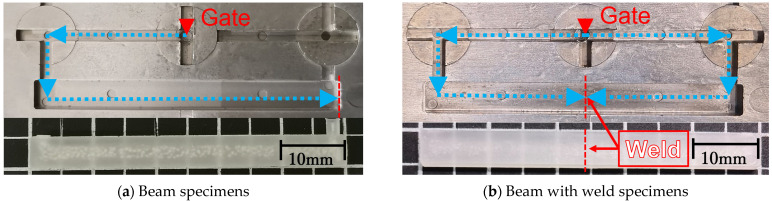
Mold cavity shapes, injection flow path, beam, beam with weld specimen’s image. Reprinted with permission from Ref. [18]. 2023. Elsevier.

**Figure 3 polymers-15-04297-f003:**
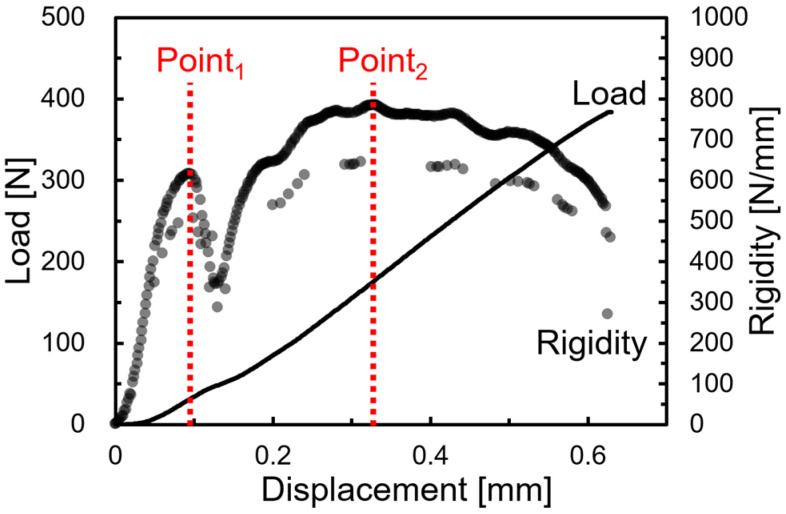
Load–displacement and rigidity–displacement curves were obtained from three-point bending tests using the short beam shear test.

**Figure 4 polymers-15-04297-f004:**
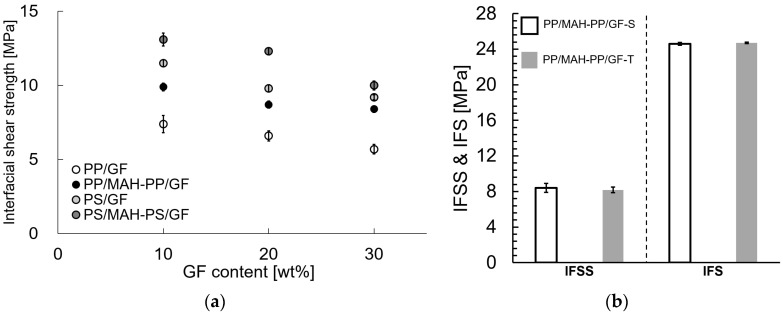
Interfacial shear strengths and interfacial strengths of different melt-mixing condition SGFRTPs. (**a**) Twin-screw extruder with 230 °C melting temperature melt-mixing condition. (**b**) Twin-screw and single-screw extruder with 200 °C melting temperature melt-mixing condition.

**Figure 5 polymers-15-04297-f005:**
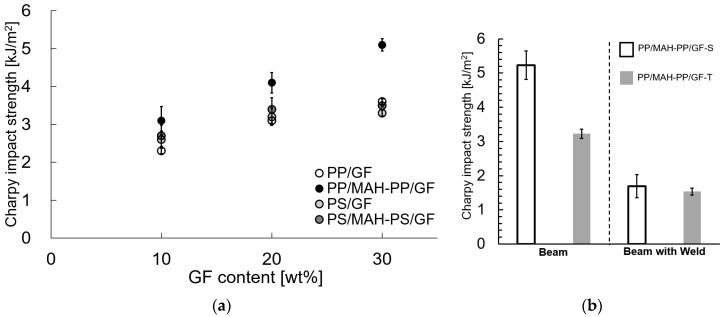
Charpy impact strengths of different melt-mixing condition SGFRTPs. (**a**) Twin-screw extruder with 230 °C melting temperature melt-mixing condition. (**b**) Twin-screw and single-screw extruder with 200 °C melting temperature melt-mixing condition.

**Figure 6 polymers-15-04297-f006:**
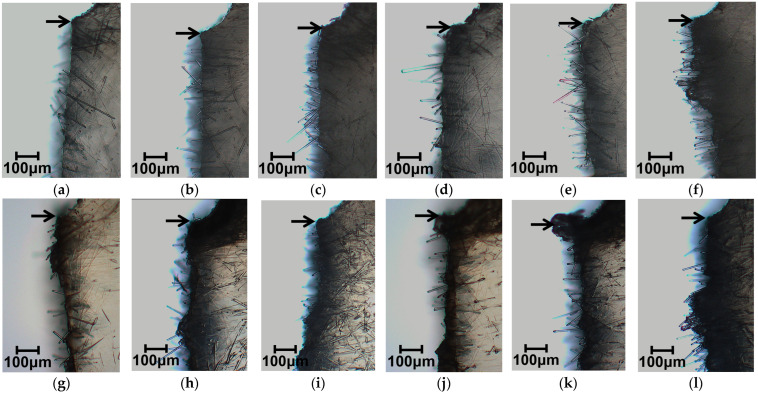
Phase contrast microscope images using the twin-screw extruder at a melting temperature of 230 °C; beam specimens’ SGFRTPs observed after Charpy impact tests. Arrows in the figures indicate the notch tip. (**a**) PP/GF-10 wt%. (**b**) PP/GF-20 wt%. (**c**) PP/GF-30 wt%. (**d**) PP/MAH-PP/GF-10 wt%. (**e**) PP/MAH-PP/GF-20 wt%. (**f**) PP/MAH-PP/GF-30 wt%. (**g**) PS/GF-10 wt%. (**h**) PS/F-20 wt%. (**i**) PS/GF-30 wt%. (**j**) PS/MAH-PS/GF-10 wt%. (**k**) PS/MAH-PS/GF-20 wt%. (**l**) PS/MAH-PS/GF-30 wt%.

**Figure 7 polymers-15-04297-f007:**
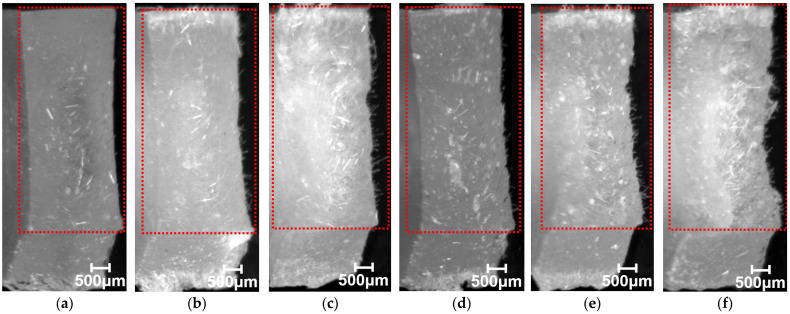
Fractured surface images using the twin-screw extruder at a melting temperature of 230 °C; beam specimens’ SGFRTPs observed after Charpy. (**a**) PP/GF-10 wt%. (**b**) PP/GF-20 wt%. (**c**) PP/GF-30 wt%. (**d**) PP/MAH-PP/GF-10 wt%. (**e**) PP/MAH-PP/GF-20 wt%. (**f**) PP/MAH-PP/GF-30 wt%. (**g**) PS/GF-10 wt%. (**h**) PS/GF-20 wt%. (**i**) PS/GF-30 wt%. (**j**) PS/MAH-PS/GF-10 wt%. (**k**) PS/MAH-PS/GF-20 wt%. (**l**) PS/MAH-PS/GF-30 wt%.

**Figure 8 polymers-15-04297-f008:**
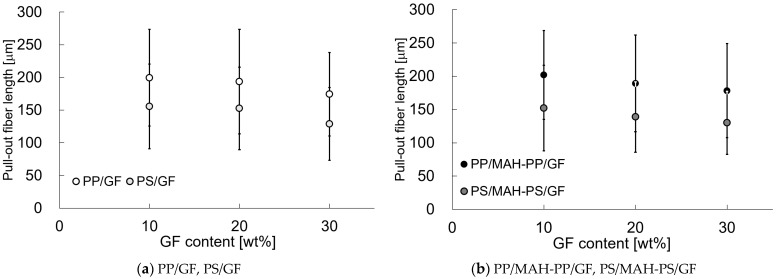
Pull-out fiber lengths of beam specimens’ SGFRTPs observed from Charpy impact tests.

**Figure 9 polymers-15-04297-f009:**
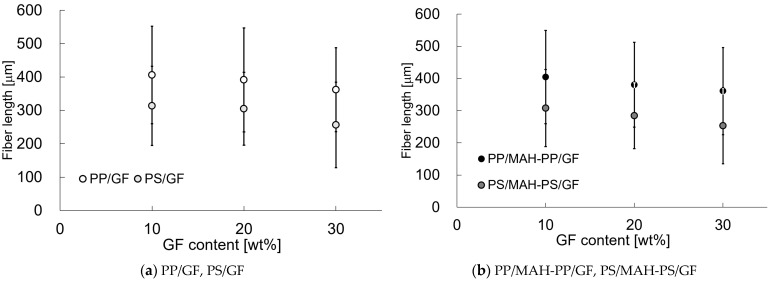
Fiber lengths of beam specimens’ injection molded SGFRTPs.

**Figure 10 polymers-15-04297-f010:**
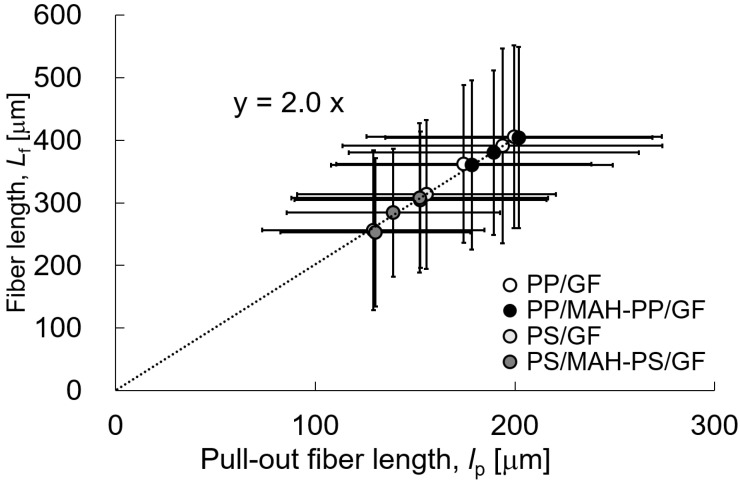
Relations between pull-out fiber length and fiber length of beam specimens’ SGFRTPs.

**Figure 11 polymers-15-04297-f011:**
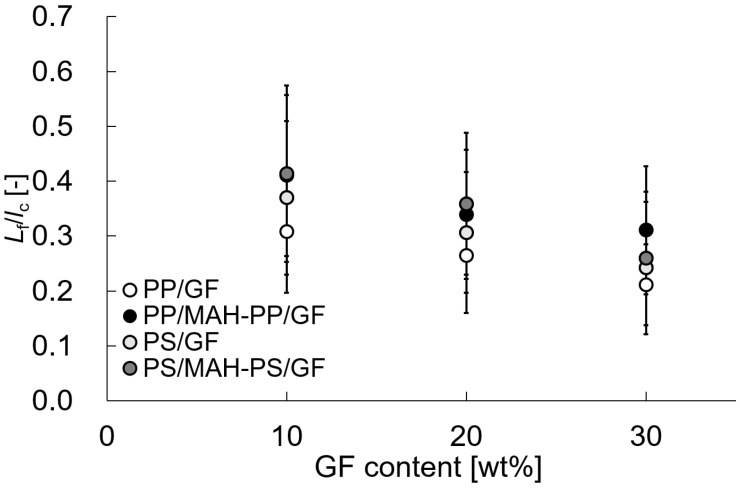
*L_f_*/*l_c_* of beam specimens’ SGFRTPs.

**Figure 12 polymers-15-04297-f012:**
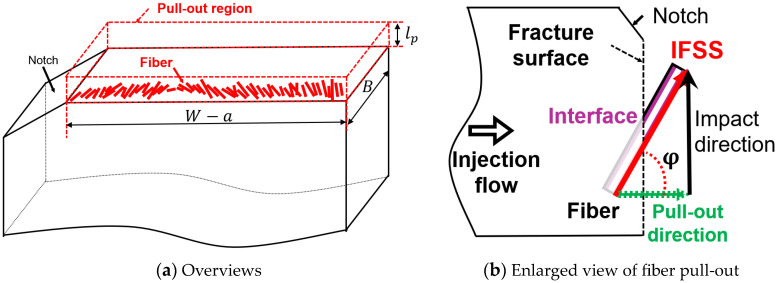
Pull-out model with an explanation of the notched impact strength. Reprinted with permission from Ref. [18]. 2023. Elsevier.

**Figure 13 polymers-15-04297-f013:**
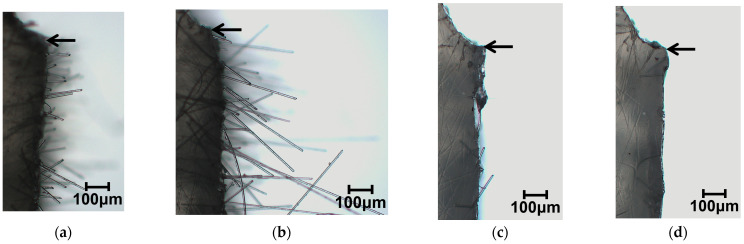
Phase contrast microscope images of using twin-screw and single-screw extruders at a melting temperature of 200 °C on beam and beam with weld specimens’ SGFRTPs after Charpy impact tests. Arrows in the figures indicate the notch tip. (**a**) PP/MAH-PP/GF-T Beam. (**b**) PP/MAH-PP/GF-S Beam. (**c**) PP/MAH-PP/GF-T Beam with Weld. (**d**) PP/MAH-PP/GF-S Beam with Weld.

**Figure 14 polymers-15-04297-f014:**
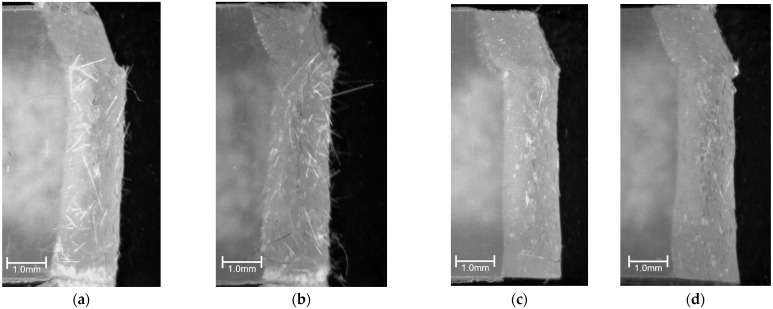
Fractured surface images using twin-screw and single-screw extruders at a melting temperature of 200 °C on beam and beam with weld specimens’ SGFRTPs after Charpy impact tests. (**a**) PP/MAH-PP/GF-T Beam. (**b**) PP/MAH-PP/GF-S Beam. (**c**) PP/MAH-PP/GF-T Beam with Weld. (**d**) PP/MAH-PP/GF-S Beam with Weld.

**Figure 15 polymers-15-04297-f015:**
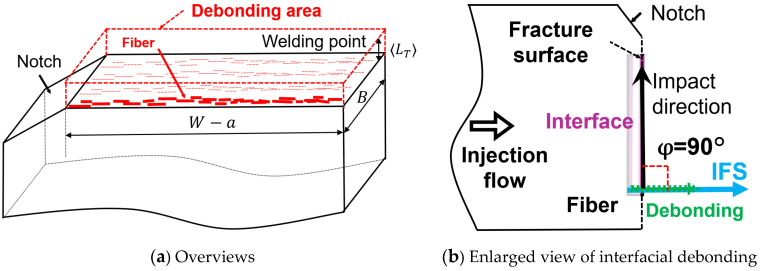
Interfacial debonding model with an explanation of the notched impact strength.

**Figure 16 polymers-15-04297-f016:**
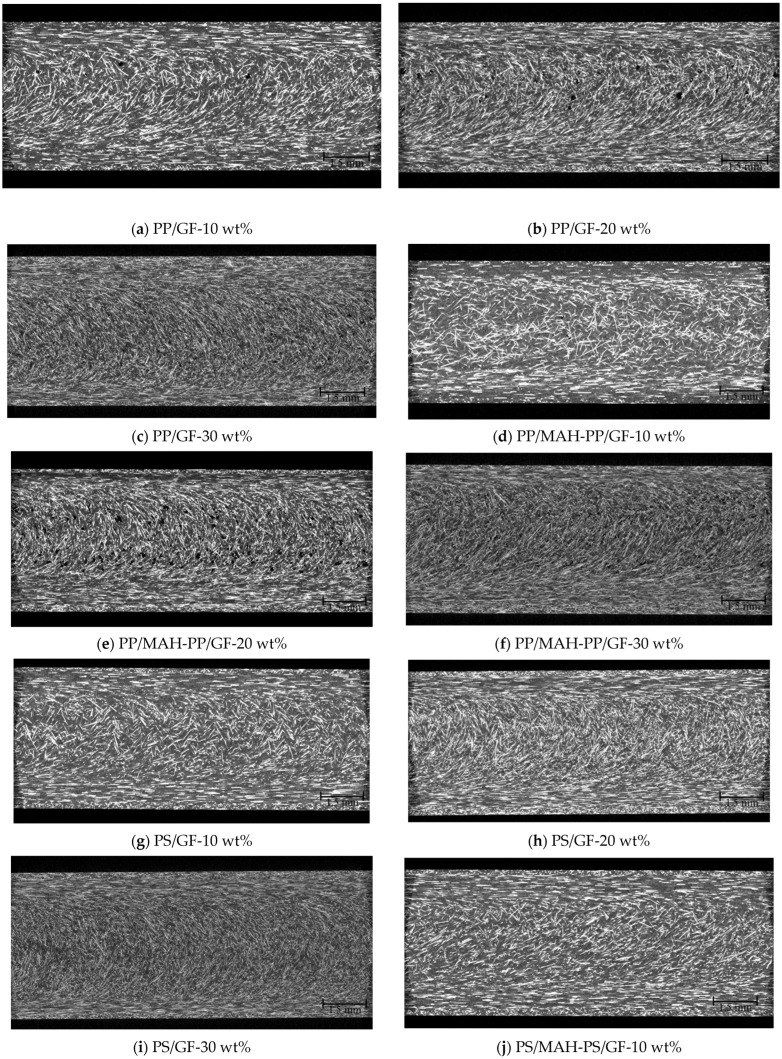
X-ray CT images using the twin-screw extruder at a melting temperature of 230 °C on injection molded beam specimens’ SGFRTPs. Reprinted with permission from Ref. [18]. 2023. Elsevier.

**Figure 17 polymers-15-04297-f017:**
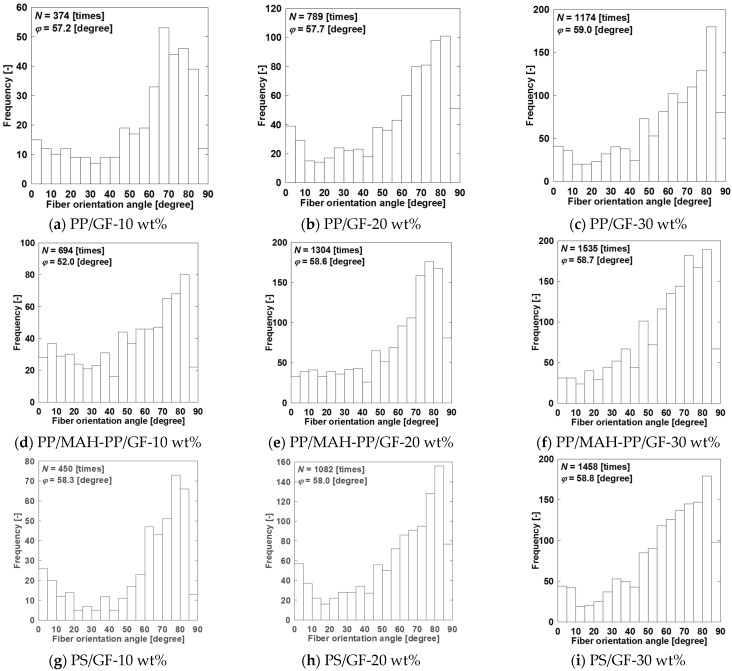
Orientation angle distributions and average orientation angles of twin-screw and single-screw extruders at a melting temperature of 200 °C on beam and beam with weld specimens’ SGFRTP.

**Figure 18 polymers-15-04297-f018:**
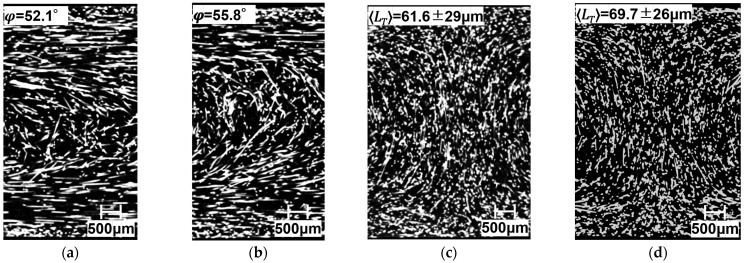
X-ray CT images, average orientation angles, and average fiber-to-fiber distance using twin-screw and single-screw extruders at a melting temperature of 200 °C on beam and beam with weld specimens’ SGFRTPs. (**a**) PP/MAH-PP/GF-T Beam. (**b**) PP/MAH-PP/GF-S Beam. (**c**) PP/MAH-PP/GF-T Beam with Weld. (**d**) PP/MAH-PP/GF-S Beam with Weld.

**Figure 19 polymers-15-04297-f019:**
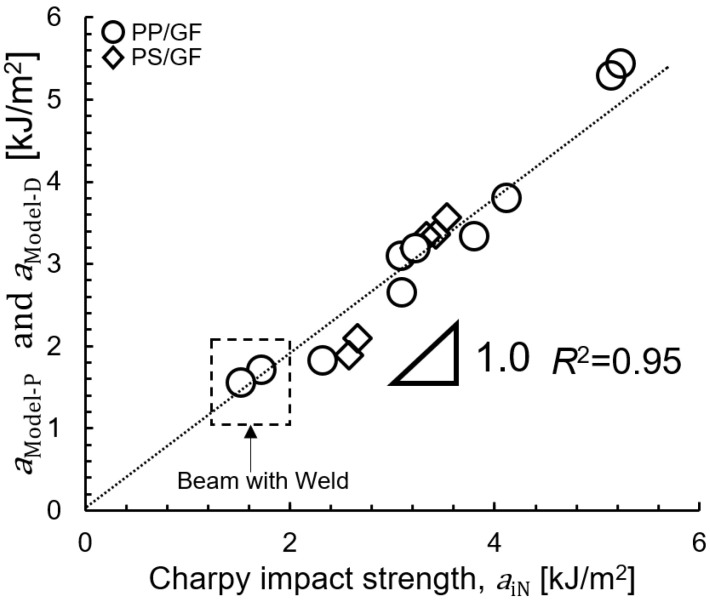
Correlation between *a_iN_*, *a_Model-P_*, and *a_Model-D_*.

**Table 1 polymers-15-04297-t001:** Factors affecting SFRTP strength and impact strength.

SFRTP Structure	Factors
Matrix	Yield stress, Elastic modulus, Poisson’s ratio, Density, Linear expansion coefficient, etc.
Fiber	Strength, Elastic modulus, Poisson’s ratio, Density, Linear expansion coefficient, etc.
Matrix/Fiber interface	Interfacial shear strength (IFSS), Interfacial strength (IFS), Friction coefficient, Interaction force, Specific surface area, etc.

**Table 2 polymers-15-04297-t002:** Mixing ratios for melt-mixing.

Code	Mixing Temp. [°C]	PP [wt%]	MAH-PP [wt%]	PS [wt%]	MAH-PS [wt%]	GF [wt%]
PP/GF	230	70				30
PP/MAH-PP/GF		68.5	1.5			30
PS/GF				70		30
PS/MAH-PS/GF				68.5	1.5	30
PP/MAH-PP/GF-S	200	85	5			10
PP/MAH-PP/GF-T		85	5			10

**Table 3 polymers-15-04297-t003:** Injection molding conditions.

Parameter	PP/GF PP/MAH-PP/GF	PS/GF PS/MAH-PS/GF	PP/MAH-PP/GF-S PP/MAH-PP/GF-T	
Specimen shape	Beam	Beam	Beam	Beam with weld
Injection temp. [°C]	230	230	200	
Mold temp. [°C]	50	50	50	
Injection speed [mm/s]	30	20	10	
Holding pressure [MPa]	84	92	84	
Injection time [s]	45	45	45	
Cooling time [s]	15	15	15	
GF content [wt%]	10, 20, 30	10, 20, 30	10	

## Data Availability

The data presented in this study are available on request from the corresponding author.

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
