# Peer review of "Impact Energy Dissipation and Quantitative Models of Injection Molded Short Fiber-Reinforced Thermoplastics"

_polymers, 2023, doi:10.3390/polym15214297_

Round 1

Reviewer 1 Report

Comments and Suggestions for Authors

The article uses PP and PS as matrix materials, MAHPP and MAHPS as additives, and adds different proportions of chopped glass fibers as experimental variables. The samples were molded and tested. Notch impact tests were conducted to analyze the changes in shear strength, interface strength and impact strength with fiber content. Finally, an energy dissipation model was obtained. Some issues are given as below.

1. What is the engineering object? What problem is studied to solve?

2. The 5% mass fraction of MAH-PS and MAH-PP in L102 is not written in Table 2. The terms  MAHPP and MAHPS in the article should be MAH-PP and MAH-PS.

3. Line 119. Why did you choose 10%wt glass fiber as a control experiment for single and twin-screw extruders instead of 20% or 30%wt glass fiber?

4. Some data in Figures 4 and 5 have no error bars.

5. Which conditions can the impact energy consumption model be used in?

Author Response

Thank you to the editors and reviewers for valuable comments. I think I was able to make the paper more understandable. The response to the comments is specified below. Corrections to the text are indicated with a yellow highlighter.

  1. What is the engineering object? What problem is studied to solve?

The engineering object is to propose a simple and versatile model for calculate impact strength of Glass short fiber reinforced thermoplastics. The solved problem is the inability to thoroughly verify the validity of the current pultrusion model and its inadequacy in accounting for the impact resistance at a fiber orientation angle of 90°. The new explanation was added in Introduction. (Line 108 to 111.)

  1. The 5% mass fraction of MAH-PS and MAH-PP in L102 is not written in Table 2. The terms MAHPP and MAHPS in the article should be MAH-PP and MAH-PS.

It's exactly as reviewer’s indicated, we do lack the 1.5% mass fraction of MAH-PS and MAH-PP in (Line 138). The terms MAHPP and MAHPS in the article was corrected. (Line 137 to Line 142.)

  1. Line 119. Why did you choose 10%wt glass fiber as a control experiment for single and twin-screw extruders instead of 20% or 30%wt glass fiber?

It's exactly as reviewer’s indicated, it would be more reasonable to use a control experiment of 20% or 30%wt glass fiber. Unfortunately, the single screw extruder utilized in this research only took into account 10 wt.% glass fiber due to the wear-prone material of the screw.

  1. Some data in Figures 4 and 5 have no error bars.

It's exactly as reviewer’s indicated, we have revised the color of the markings to make the range of error bars clearly visible (Figure 4 and Figure 5).

  1. Which conditions can the impact energy consumption model be used in?

It's exactly as reviewer’s indicated, We lack a description of the scope of application of the model. As explained in discussion 4.4 (Line 373 to 378), model can be used when fiber pull-out or interfacial debonding is the main energy dissipation mechanism. According to the experimental results, we think that the interfacial debonding model is suitable when the fiber orientation angle of the short fiber injection molding product is all parallel to the impact direction. This is a very special state of fiber dispersion. In most cases, the impact strength prediction is suitable for the fiber pull-out model. This explains is added in (Line 339 to 341)

The manuscript has checked by a colleague fluent in English writing.

Reviewer 2 Report

Comments and Suggestions for Authors

In general, the article is scientific. A quantitative model was clarified to calculate the IFSS, IFS and impact strength. However, the title is the impact energy dissipation mechanism of materials, there are no explain in the article to show the energy dissipation. At the same time, some references do not correspond to the content of the article.

Some comments and suggestions that the authors may want to address.

Comment 1: In line 10, what does “strength” stand for? In my opinion, the strength includes impact resistance.

Comment 2: In line 12 and 13, it was said that the model shows good agreement with experiment. Please show specific data to prove it, including the maximum error and coefficient of determination.

Comment 3: In line 21 and 22, there are many ways to emit greenhouse gases. Why do you believe that the reduction of emission from transportation equipment is particularly required? Please prove it by references.

Comment 4: In the introduction, I would recommend to analyze the existing studies in more details. Now it looks very brief, therefore, missing important points of the justification of the current research.

Comment 5: In the 2.1., what is MFR and Vicat Softening Temperature? Please add more details of the materials including matrices, fibers and additives.

Comment 6: Whether the impact specimens meet to any standards? Please indicate these standards or show the specific size of the impact specimens.

Comment 7: For equation (4), the orientation angle calculation formula may false. The orientation angle I calculated is arctan(P2/P1). Please check the formula and make sure there are no similar question in the article.

Comment 8: In line 158, Jiang’s article was referenced, however, the article at the references list does not match. Please check it.

Comment 9: Part 2.5. and 2.6. The whole paragraph references the word of other researchers, which is inappropriate. Please indicate the specific reference location.

Comment 10: The error bar in figure 8 and figure 9 are unclearly. I cannot determine whit kind of material each label is out of error.

Comment 11: Please explain the impact energy dissipation mechanism. What is the different from it to the impact mechanism? And how do you investigate the dissipation in your article.

Comment 12: In the conclusion, the results should be broken down to several points. And specific data of the investigation should also be outlined to show the novelty of the article.

Comments on the Quality of English Language

Some prepositions need further refinement.

The subject of many sentences is too long, which makes understanding difficult.

Author Response

Thank you to the editors and reviewers for valuable comments. I think I was able to make the paper more understandable. The response to the comments is specified below. Corrections to the text are indicated with a yellow highlighter.

Comment 1: In line 10, what does “strength” stand for? In my opinion, the strength includes impact resistance.

It's exactly as reviewer’s indicated, In the (Line 10), the definition of strength is not clear. Here we want to clarify the scope of mechanical properties. The new explanation was added in Abstract. (Line 9 to 10.)

Comment 2: In line 12 and 13, it was said that the model shows good agreement with experiment. Please show specific data to prove it, including the maximum error and coefficient of determination.

It's exactly as reviewer’s indicated, we do lack on the maximum error and coefficient of determination. The new explanation was added in Abstract. (Line 14.)

Comment 3: In line 21 and 22, there are many ways to emit greenhouse gases. Why do you believe that the reduction of emission from transportation equipment is particularly required? Please prove it by references.

Fuel consumed by transportation is one of the main sources of greenhouse gases. The improvement of fuel efficiency for transportation equipment is a very important measure to curb greenhouse gas emissions. Relevant references have been added at (Line 24.) (References [1]).

Comment 4: In the introduction, I would recommend to analyze the existing studies in more details. Now it looks very brief, therefore, missing important points of the justification of the current research.

It's exactly as reviewer’s indicated, In the introduction the existing studies do lack details. In particular, the explanation on the relationship between the existing impact test and interfacial shear strength is not enough. The new explanation was added in Introduction. (Line 76 to 108.)

Comment 5: In the 2.1., what is MFR and Vicat Softening Temperature? Please add more details of the materials including matrices, fibers and additives.

It's exactly as reviewer’s indicated, we do lack on the materials including matrices, fibers and additives. The details of the materials was added in 2.1. Materials. (Line 118 to 121.)

Comment 6: Whether the impact specimens meet to any standards? Please indicate these standards or show the specific size of the impact specimens.

It's exactly as reviewer’s indicated, we do lack the standards or show the specific size of the impact specimens. The details of the test samples were added in 2.2. Sample Preparations. (Line 149.)

Comment 7: For equation (4), the orientation angle calculation formula may false. The orientation angle I calculated is arctan(P2/P1). Please check the formula and make sure there are no similar question in the article.

It's exactly as reviewer’s indicated, in equation (4) the orientation angle calculation formula was false. I have fixed the equation. Thank very much for carefully check. We checked all the equations carefully to make sure there was no problem with the calculation.

Comment 8: In line 158, Jiang’s article was referenced, however, the article at the references list does not match. Please check it.

It's exactly as reviewer’s indicated, the reference number in 2.4. Short Beam Share Testing for Determination of IFSS. I have rechecked and revised the serial number of the reference. (Line 195.).

Comment 9: Part 2.5. and 2.6. The whole paragraph references the word of other researchers, which is inappropriate. Please indicate the specific reference location.

It's exactly as reviewer’s indicated, the whole paragraph references the word of other researchers, which is inappropriate. I have deleted the reference number of Part 2.5. and 2.6. I have rechecked and revised the specific reference location (Line 204.).

Comment 10: The error bar in figure 8 and figure 9 are unclearly. I cannot determine whit kind of material each label is out of error.

It's exactly as reviewer’s indicated, the error bar in figure 8 and figure 9 are unclearly. We have split the icons to present them clearly (Figure 8 and Figure 9).

Comment 11: Please explain the impact energy dissipation mechanism. What is the different from it to the impact mechanism? And how do you investigate the dissipation in your article.

It's exactly as reviewer’s indicated, the description used in the article is easy to cause misunderstanding. The impact mechanism was the same meaning as fracture mechanism. The main purpose of this study is to purpose a quantitative model related to impact strength. I replaced the words ‘Impact Mechanism’ that are prone to misunderstanding into ‘Energy Dissipation Quantitative Models’ thought the article. (Line 2, Line 261, Line 300, Line 304, Line 327, Line 378, Line 379, Line 396, Line 404, and Line 409.)

Comment 12: In the conclusion, the results should be broken down to several points. And specific data of the investigation should also be outlined to show the novelty of the article.

It's exactly as reviewer’s indicated, the specific data of the survey should be summarized in the conclusion to show the novelty of the article. The specific data was added in conclusion. (Line 393 to 406.)

The manuscript has checked by a colleague fluent in English writing.

Reviewer 3 Report

Comments and Suggestions for Authors

The author presented an interesting study on a quantitative correlation between impact stress and interfacial shear stress, with sound rationale and enough experiment evidence. However, it’s an extension of a previous work, which limited its innovation. I suggest acceptance of the paper after some major revision.

1.       In the end of the introduction part, the author should state more on the significance of this study. The positive correlation between IFSS and impact strength is intuitive and well known. How a qualitative model can be beneficial to the community? Can it be a general model that people can easily use for different molded parts? Or is it a conditional model depending on different matrix and fillers? Which factors can be dominant and which not, any literature reviews on this?

2.       The author showed the results of both w&wo MA. Then the discussion on MA’s impact should also be included through the whole discussion. i.e. regarding figure.4 and 5, rather than simply referring the literature, the author should simply explain the impact of MA on the enhanced mechanical properties of the composite. Regarding figure 8 to 11, why some of the result w/wo MA show limited difference (fig.8,9) while some showing difference (fig.11)? MA or other possible additive enhances the surface affinity between the matrix and the gf. However, are they really a factor to be involved in some correlations established in this work (i.e fig. 10 and 19)?

3.       In Fig12, are you wanna show the fibers perpendicular to the cross-section? Fiber orientation is not clear the cartoon. And better to be redraw or add a view of another side.

4.       From Fig19, the conclusion seems like the fiber orientation at the interface wont affect the universality of the model. However, I didn’t see any clear conclusion on it in the discussion part.

5.       In fig 19. The two groups (PS, PP) can probably be fitted individually. Can you obtain higher R2 for each by doing so? Does it means the model could be matrix dependent?

Comments on the Quality of English Language

na

Author Response

Thank you to the editors and reviewers for valuable comments. Thanks to you, I think I was able to make the paper more understandable. The response to the comments is specified below. Corrections to the text are indicated with a yellow highlighter.

  1. In the end of the introduction part, the author should state more on the significance of this study. The positive correlation between IFSS and impact strength is intuitive and well known. How a qualitative model can be beneficial to the community? Can it be a general model that people can easily use for different molded parts? Or is it a conditional model depending on different matrix and fillers? Which factors can be dominant and which not, any literature reviews on this?

It's exactly as reviewer’s indicated, we do lack on the research significance of this research. We have attempted to explain from a larger perspective the factors that affect the mechanical properties of SFRTP (Table 1). Table 1 shows factors affecting the SFRTP structure including Matrix/ Fiber interface. At the same time, we have listed several papers to illustrate the positive correlation between IFSS and impact strength is intuitive and well known (Line 74 to 75.). But a quantitative model expressing impact strength using IFSS and IFS has not been proposed. We hope that the results of this study can provide a simple and universal calculation method. The new explanation was added in introduction. (Line 102 to 108.)

  1. The author showed the results of both w&wo MA. Then the discussion on MA’s impact should also be included through the whole discussion. i.e. regarding figure.4 and 5, rather than simply referring the literature, the author should simply explain the impact of MA on the enhanced mechanical properties of the composite. Regarding figure 8 to 11, why some of the result w/wo MA show limited difference (fig.8,9) while some showing difference (fig.11)? MA or other possible additive enhances the surface affinity between the matrix and the gf. However, are they really a factor to be involved in some correlations established in this work (i.e fig. 10 and 19)?

It's exactly as reviewer’s indicated, we do lack discussion on the effect of MA addition. The new explanation was added in discussion. (Line 280 to 285 and Line 297 to 299.) We would like to emphasize that the model proposed in this paper can also be applied to both w&wo MA.

  1. In Fig12, are you wanna show the fibers perpendicular to the cross-section? Fiber orientation is not clear the cartoon. And better to be redraw or add a view of another side.

It's exactly as reviewer’s indicated, the fiber orientation is not clear the cartoon. What we want to demonstrate is that the fibers form a certain random angle at the fracture surface, like phase contrast microscope observation results (figure 6). To avoid misunderstandings, we have added a flat explanation image (figure 12(b) and figure 15(b)).

  1. From Fig19, the conclusion seems like the fiber orientation at the interface wont affect the universality of the model. However, I didn’t see any clear conclusion on it in the discussion part.

It's exactly as reviewer’s indicated, the fiber orientation at the interface won’t affect the universality of the model. We do lack a description of it in the conclusion. The new explanation was added in conclusion. (Line 407 to Line 408.)

  1. In fig 19. The two groups (PS, PP) can probably be fitted individually. Can you obtain higher R2 for each by doing so? Does it means the model could be matrix dependent?

It's exactly as reviewer’s indicated, the original R2 results were relatively low. We believe that this is due to the impact of the fiber length measurement results on the accuracy of the calculation. So, we increased the number of samples for measuring fiber length and pull-out fiber length. Fiber length has increased to over 500, and pull-out fiber length has increased to over 300. And the new explanation was added of clarified the measurement quantity method. (Line 214 and Line 229.) We have also updated (Figure 19) that obtain higher R2. We would like to emphasize that the model is suitable for different matrices.

The manuscript has checked by a colleague fluent in English writing.

Round 2

Reviewer 1 Report

Comments and Suggestions for Authors

The manuscript can be accepted now. 

Reviewer 3 Report

Comments and Suggestions for Authors

this draft is recommended to be accepted